# Advancing Pharyngeal Constrictor Muscle Auto-Segmentation: A Comparative Analysis

**Anju Kaimal**[1]                                            ANJU.KAIMAL@ICR.AC.UK

**Konstantinos Zormpas-Petridis**[2]                          KONSTANTI-

NOS.ZORMPASPETRIDIS@POLICLINICOGEMELLI.IT

**Justine Tyler**[3]                                          JUSTINE.TYLER@RMH.NHS.UK

**Christopher M. Nutting**[3]                                 CHRIS.NUTTING@RMH.NHS.UK

**Matthew D. Blackledge**[1]                                  MATTHEW.BLACKLEDGE@ICR.AC.UK

[1] *Institute of Cancer Research, London, United Kingdom*

[2] *Fondazione Policlinico Universitario Agostino Gemelli IRCCS, Rome, Italy*

[3] *The Royal Marsden NHS Foundation Trust, London, United Kingdom*

**Editors:** Under Review for MIDL 2024

## Abstract

Accurate auto-segmentation of the pharyngeal constrictor muscle (PCM) is crucial for precise treatment planning in head and neck cancer. This study compares three deep-learning segmentation methods: (i) 2D U-Net, (ii) 3D UNet with data fingerprinting (nnUNet), and (iii) our proposed 2D statistical curve fitting method using transfer learning. Using a total of 168 planning CT images for training and validation, results indicate varying effectiveness, with the nnUNet exhibiting the lowest mean surface distance (2.73mm, inter-quartile range = 1.02mm), statistical curve fitting obtained a mean-surface distance of 8.25 mm with IQR of 6.33 mm. Though this was not as accurate as nnUNet, but was significantly superior than a conventional 2D UNet model.

**Keywords:** Auto-contouring, Head and neck cancer, radiotherapy planning

## 1. Introduction

Delineation of the pharyngeal constrictor muscles (PCM) is crucial for precise radiotherapy treatment planning in head and neck cancer. Manual delineation, though commonly employed in clinical practice, can be prone to inter-observer variability and is time-consuming (Petkar et al., 2016, 2017). Deep learning techniques show promise in addressing these challenges(Nutting et al., 2023; Van Dijk et al., 2020), though automatic delineation of the PCM remains challenging as they are often not clear on images. This paper compares three methodologies for PCM segmentation: a 2D convolutional approach using a conventional U-Net(Ronneberger et al., 2015) with a dice loss function, a 3D U-Net approach integrated into nnUNet with data fingerprinting (Isensee et al., 2021), and our suggested deep-learning statistical curve fitting approach, which models the PCM on each CT image slice as a parametric curve consisting of a number of ordered pixel-coordinates.

## 2. Methodology

### 2.1. Dataset Description

The study employs a dataset comprising 168 planning CT images acquired from individuals diagnosed with head and neck cancer at a single center. All images included clinically annotated contours of the PCM for treatment planning purposes, which provided the gold-standard for training and testing. Out of the data, 34 patients were kept for testing, and the remaining data were randomly split using a 80:20 ratio for training and validation.

### 2.2. Model architectures

**2D UNet** We converted single CT images into size $256 \times 256 \times 3$ by replicating the image across all three channels. The 2D UNet architecture was initiated using ImageNet weights, and trained using a dice loss, minimized using stochastic gradient descent with a learning rate of 0.01 over 500 epochs.

**3D nnUNet** The nnUNet combines the 3D UNet architecture with data fingerprinting techniques to enhance segmentation accuracy. Data fingerprinting emphasizes significant features whilst reducing noise, and improving segmentation quality. Trained for 1000 epochs with automatic CT image normalization, nnUNet streamlines preprocessing, achieving superior segmentation performance. (Isensee et al., 2021).

**Statistical Curve Fitting** We introduce a novel approach to 2D contouring of the PCM by approximating it's shape as parametric polynomial curve. We fit a polynomial curve to the clinically delineated masks within each CT image such that they may be described as a parametric polynomial of the form $((x_1, y_1), \ldots, (x_N, y_N))$ where $x_i = a_0 + a_1 u_i + a_2 u_i^2 + \ldots a_{m_x} u_i^{m_x}$ and $y_i = b_0 + b_1 u_i + b_2 u_i^2 + \ldots b_{m_y} u_i^{m_y}$, with $u = (0, 1/(N-1), 2/(N-1), \ldots, 1)$ (see Figure 1a). Coefficients $a_j$ and $b_j$ are fit to training masks using a previously proposed method (par), with $m_x = m_y = 3$ and subsequently sampling $N = 10$ equidistant vertices. We train two fully-connected models: (i) a binary classification model with softmax output layer for determining the presence of PCM within a CT slice, and (ii) a regression model with linear output layer for predicting the contour vertices $(x_i, y_i)$. Input to these models consisted of comparing features derived from 47 existing model architectures within models within TensorFlow, with pre-trained weights derived from the ImageNet database. Input to pre-trained models consisted of three channels comprising the CT image (red channel), column number (green channel), and row number (blue channel). Both fully connected networks were trained for 500 epochs using mean square error loss for the regression network and binary cross entropy for the classification network. The classification model consisted of 3 hidden, flat layers of size 128, 64 and 32 respectively (ReLu activation), and the regression model consisted of 2 hidden layers, both with size 64 (ReLu activation).

Mean Surface Distance (MSD) and Hausdorff Distance (HD) were used as a validation metrics to compare the accuracy of all model architecture tested. The performance of these methodologies was evaluated against a previously published benchmark (Petkar et al., 2017), using data from the same study to assess inter-observer variability in manual delineation of the PCM.

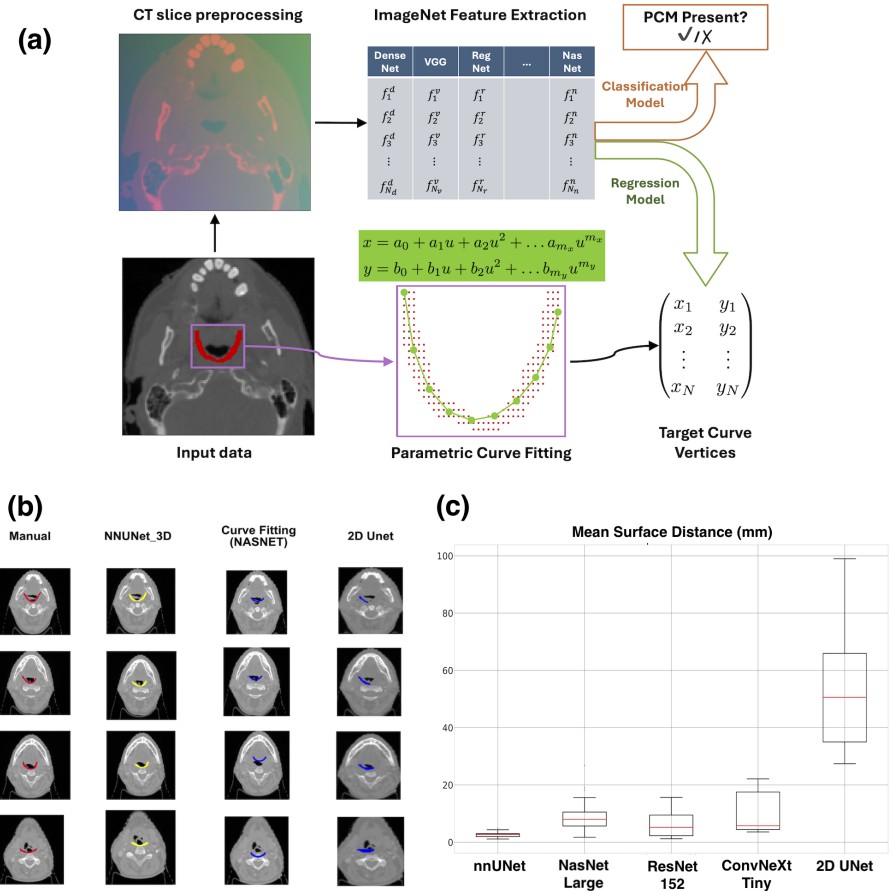

Figure 1: (a) Illustration of our curve-fitting deep-learning pipeline for delineation of the PCM. (b) Example results for evaluated techniques.(c) Performance comparison between nnUNet, 2D UNet, and top 3 curve-fitting models.

## 3. Results & Discussion

Figure 1(b) displays exemplar test CT images, along with segmentation results from the various model architecture we tried. Curve fitting methods with ImageNet backbones like NasNetLarge and ResNetRS152 outperform 2D UNet but still fall short of 3D nnUNet. Figure 1(c) shows varying segmentation effectiveness, with nnUNet achieving the best precision (median MSD: 2.7 mm [IQR: 1.0 mm], median HD: 6.4 mm [IQR: 4.9 mm]). NasNet-based Statistical Curve Fitting, though slightly inferior (median MSD: 8.3 mm [IQR: 6.3 mm], median HD: 14.0 mm [IQR: 5.8 mm]), holds promise. Further research is needed to validate curve fitting's superiority over 2D UNet and explore its extension to 3D. We anticipate that a major advantage of the statistical curve fitting method explored here is that it offers the opportunity to explore the use of shape-based priors on region segmentation (e.g. length and curvature), which could offer constraints on the results. This is especially important for PCM delineation, where target regions are not clear within the image.

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
