# OpenReview forum: "Advancing Pharyngeal Constrictor Muscle Auto-Segmentation: A Comparative Analysis"
_MIDL.io/2024/Short_Papers — MIDL 2024 Short Papers_

### Official Review · Reviewer_rwF3 · 2024-04-22

**Confidence:** 4
**Final Rating:** 3.5

**Review:**

The paper introduces a novel 2D statistical curve fitting method using transfer learning for segmenting the pharyngeal constrictor muscle (PCM) in 3D CT images, utilizing shape priors. It also compares three different segmentation methods across 168 planning CT images, which helps confirm the robustness of the findings.

Despite these strengths, there are several areas in the paper that require improvement:

1.	Clarity in Methodology: The description of how the predicted vertices are transformed into segmentation masks is unclear. Are these vertices all pixels inside the segmentation masks?

2.	Analysis of Method Effectiveness: The paper lacks a discussion on why the proposed method underperforms compared to the 3D nnUNet. This could be due to multiple factors, such as the accuracy of the classification model in detecting PCM presence or issues in the transformation process of predicted vertex locations to binary masks. A detailed analysis of these potential causes would provide valuable insights into improving the proposed method.

3.	Comprehensive Evaluation Metrics:There is a noticeable absence of other critical evaluation metrics such as the Dice score and Jaccard index.

---

### Decision · Program_Chairs · 2024-04-26

Accept